# Determination of the [^15^N]-Nitrate/[^14^N]-Nitrate Ratio in Plant Feeding Studies by GC–MS

**DOI:** 10.3390/molecules24081531

**Published:** 2019-04-18

**Authors:** Sebastian Schramm, Maria Fe Angela Comia Boco, Sarah Manzer, Oliver König, Tong Zhang, Fatima Tuz Zohora Mony, Adebimpe Nafisat Adedeji-Badmus, Brigitte Poppenberger, Wilfried Rozhon

**Affiliations:** Biotechnology of Horticultural Crops, TUM School of Life Sciences Weihenstephan, Technical University of Munich, Liesel-Beckmann-Straße 1, 85354 Freising, Germany; seb.schramm@tum.de (S.S.); mariafefeangela.boco@studio.unibo.it (M.F.A.C.B.); sarah.manzer@tum.de (S.M.); oliver.koenig@tum.de (O.K.); ge52qug@mytum.de (T.Z.); fatima.mony@tum.de (F.T.Z.M.); adebimpe.adedeji@tum.de (A.N.A.-B.); brigitte.poppenberger@wzw.tum.de (B.P.)

**Keywords:** GC–MS, mesitylene, nitrate, nitrogen, feeding experiments, stable isotope

## Abstract

Feeding experiments with stable isotopes are helpful tools for investigation of metabolic fluxes and biochemical pathways. For assessing nitrogen metabolism, the heavier nitrogen isotope, [^15^N], has been frequently used. In plants, it is usually applied in form of [^15^N]-nitrate, which is assimilated mainly in leaves. Thus, methods for quantification of the [^15^N]-nitrate/[^14^N]-nitrate ratio in leaves are useful for the planning and evaluation of feeding and pulse–chase experiments. Here we describe a simple and sensitive method for determining the [^15^N]-nitrate to [^14^N]-nitrate ratio in leaves. Leaf discs (8 mm diameter, approximately 10 mg fresh weight) were sufficient for analysis, allowing a single leaf to be sampled multiple times. Nitrate was extracted with hot water and derivatized with mesitylene in the presence of sulfuric acid to nitromesitylene. The derivatization product was analyzed by gas chromatography–mass spectrometry with electron ionization. Separation of the derivatized samples required only 6 min. The method shows excellent repeatability with intraday and interday standard deviations of less than 0.9 mol%. Using the method, we show that [^15^N]-nitrate declines in leaves of hydroponically grown *Crassocephalum crepidioides*, an African orphan crop, with a biological half-life of 4.5 days after transfer to medium containing [^14^N]-nitrate as the sole nitrogen source.

## 1. Introduction

Nitrogen represents a very important fraction of a plant’s chemical composition. It is an essential plant macro-nutrient, necessary to ensure correct development and growth. Nitrogen is a fundamental constituent of amino acids, the building blocks of peptides and proteins. It is present in chlorophyll, a crucial molecule in photosynthesis, and in the phytohormones auxin and cytokinin. In addition, nitrogen is a constituent of the nucleobases present in DNA, RNA, and in many co-substrates, for instance ATP (adenosine triphosphate), NADH (nicotinamide adenine dinucleotide), and UDP-glucose (uridine diphosphate glucose), involved in a tremendous number of metabolic processes. It is also present in polyamines, quaternary ammonium compounds [1], plant defense compounds including alkaloids [2,3,4,5], glucosinolates [6,7], benzoxazinoids [8], some phytoalexins [9], and in many other metabolites. Plants grown in nitrogen-rich substrates may contain more than 5% nitrogen in dry weight while plants grown under nitrogen-limiting conditions contain still approximately 0.5–1.0% nitrogen in dry weight [10,11]. Plants take up nitrogen via the root system mainly in form of nitrate and ammonium. In most substrates nitrate is the predominant form but ammonium may be present at high levels in flooded soils [12] and in grasslands [13]. Ammonium can be incorporated directly in the root into amino acids or under some conditions, particularly at a high supply, may be transported to the shoot [14,15]. In contrast, nitrate is mainly transported to the shoot and reduced by nitrate reductase to nitrite, which is in turn further reduced by nitrite reductase to ammonium. Ammonium is incorporated by the actions of glutamine synthetase and glutamate synthetase into glutamine and glutamate [16], respectively, which serve as precursors for biosynthesis of further amino acids [17] and other nitrogen-containing compounds. Since ammonium [18,19,20] and nitrite [21] are toxic for plant tissues, nitrate reduction is well regulated [22,23,24] to avoid accumulation of these intermediates. Consequently, nitrate may accumulate in leaves under rich fertilization.

Sufficient nitrogen supply is crucial for high plant biomass production and yield, which has been studied in many plant species including the cereals wheat [25,26], maize [27,28,29], oat [30], and barley [31,32], oilseeds like canola [25] and sunflower [33], root crops like potato [34,35] and sugar beet [36], vegetables including spinach [37], lettuce [10], and kohlrabi [11] and energy crops like willow [38] and *Miscanthus sinensis* [39]. Even the legumes, which possess root nodules that contain nitrogen-fixing symbionts, show increased yields when grown in nitrate-rich substrates [40,41]. However, fertilizers must be applied at the demand of the plants since overfertilization causes severe environmental problems including contamination of surface and groundwater with nitrate [42] and is responsible for algal bloom in the sea [43]. In addition, overfertilization of vegetables like lettuce, spinach, celery, and Chinese cabbage causes nitrate accumulation at levels of up to several g/kg fresh weight [44,45,46,47]. Possible health impacts of a high dietary intake of nitrate are intensively and controversially discussed [48,49,50,51,52,53,54,55].

Due to the importance of nitrate, a number of methods have been developed for quantification of nitrate in soils, growth substrates and samples from plants, animals, and humans. These methods include spectrophotometry [56,57], ion sensitive electrode [58], ion chromatography [59], ion-pair chromatography [60], gas chromatography [61], and capillary electrophoresis [62]. In addition, soluble nitrogen-containing compounds, particularly the amino acids, can be quantified in plant tissues for assessing the nutritional status of plants and ripening of fruits [63,64,65,66]. While these methods are useful for evaluating plant nutritional status, they are of limited use for investigation of nitrogen fluxes in plants. Metabolic fluxes are often investigated by so called pulse–chase analysis, where organisms or cells are initially incubated with a labeled compound (the pulse), which will be incorporated into the metabolites. Subsequently, the labeled compound is removed and replaced by the unlabeled version. Depletion of the labeled molecule and its conversion to labeled metabolites is measured over time during the so called ‘chase’ period. Due to their simple detection, classically radio isotopes were used for such experiments. In case of nitrogen, the radionuclide ^13^N has been used for tracer studies [14,67,68,69]. However, the short ^13^N half-life of 9.97 min [70] is a drawback since ^13^N must be produced on site. Alternatives to radionuclides include stable isotopes, which also allow for performing field studies. The two stable isotopes of nitrogen are ^14^N, which has an abundance in the atmosphere of 99.634 mol%, and ^15^N with an atmospheric abundance of only 0.366 mol% [71]. The huge difference in abundance makes ^15^N a good tracer for metabolic studies. This technique has, for instance, been used to quantify nitrogen fixation [72], estimate the nitrogen flow in alfalfa plants [73], investigate nitrogen assimilation and amino acid translocation in barley [68,74,75], follow nitrogen remobilization from vegetative organs in pea [76], study nitrogen metabolism in buds of *Picea glauca* [77], and observe nitrogen fluxes in *Arabidopsis thaliana* seeds during the initiation of germination [78]. ^15^N labeling is also useful for obtaining metabolites and proteins [79] accessible for NMR spectroscopy since the ^15^N nucleus gives much sharper signals than the ^14^N nucleus [77].

^15^N contents can be analyzed by elemental analysis–isotope ratio mass spectrometry (EA–IRMS) [80,81], gas chromatography coupled to a combustion and reduction unit and a mass spectrometer [82], and by nuclear magnetic resonance spectroscopy (NMR) [77,78,83,84,85]. Other methods, including emission spectroscopy [86] and IR spectroscopy [87,88], are rarely used.

For feeding experiments with stable nitrogen isotopes [^15^N]-potassium nitrate is most commonly used. For designing an appropriate experimental setup, the kinetics of replacement of [^14^N]-nitrate by [^15^N]-nitrate (or vice versa) in the plant tissue are relevant. The methods mentioned above measure either the total nitrogen content (EA-IRMS, emission spectroscopy), cannot analyze nitrate (nitrate is non-volatile and cannot be analyzed directly by GC), or lack insensitivity for analysis of small sample quantities (NMR) and are thus not suitable for rapid determination of the ratio of [^15^N]-nitrate/[^14^N]-nitrate in limited amounts of plant material.

Several methods for quantification of nitrate in biological samples, mainly of human or animal origin, using [^15^N]-nitrate as an internal standard have been described. Green et al. presented a method for quantification of nitrate by spiking the sample with [^15^N]-nitrate followed by reaction with benzene to nitrobenzene. The ratio of [^14^N] and [^15^N]-nitrobenzene was subsequently analyzed by gas chromatography–mass spectrometry (GC–MS) to determine the nitrate concentration [89]. Similar procedures using other, more reactive aromatic compounds were also developed [90,91,92]. Other methods made use of reaction of nitrate with pentafluorobenzyl bromide and subsequent GC–MS analysis of the obtained derivatives [93,94]. Here we tested whether these methods can be applied for determining the [^15^N]-nitrate/[^14^N]-nitrate ratio in plant material using GC–MS with electron impact ionization. We found that mesitylene gives the best results as the derivatization reagent. We optimized all the steps of the method including extraction, derivatization, and detection. Possible interferences with plant metabolites were investigated and finally the method was validated. The method presented can be a helpful tool for establishing appropriate conditions for ^15^N feeding and pulse–chase experiments for measuring the level of stable isotope labeled nitrate in plants.

## 2. Results

Several GC–MS-based methods for quantification of nitrate in mammalian fluids, mainly urine, saliva, and plasma, have been reported. They depend on reaction of the nitrate ion with 2,3,4,5,6-pentafluorobenzyl bromide [93,94,95,96], benzene [89], 1,3,5-trimethoxybenzene [92], or mesitylene [90,91] to form organic compounds, which can be sensitively analyzed by GC–MS. 1,2,3,4,5-Pentafluorobenzyl bromide reacts with nucleophiles under mild conditions. Although nitrate is a poor nucleophile, it can substitute the bromine of 1,2,3,4,5-pentafluorobenzyl bromide to form a nitric acid ester. The other compounds mentioned above are nitrated via electrophilic aromatic substitution: nitrate reacts first with sulfuric acid to the nitronium cation, which subsequently combines with the aromatic ring to yield the nitrated aromatic compounds (Figure 1A). In that case nitrogen is directly linked to carbon and the obtained derivatives are very stable, which makes them highly suitable for quantification. The reaction is very specific for nitrate; the only side reaction that may happen is sulfonation of the aromatic reagent. Since the nitro group has a strong inactivating effect on the aromatic system, double derivatization is not observed. Among the reagents mentioned above, 1,3,5-trimethoxybenzene has, due to the activating effect of the methoxy groups, the highest reactivity. Mesitylene is, because of the methyl groups, slightly activated while benzene shows the lowest reactivity among these reagents.

To test the suitability of these reagents for determination of the [^15^N]/[^14^N]-nitrate ratio in plant leaves by GC–EI–MS (gas chromatography–electron ionization–mass spectrometry), test reactions were performed. Reaction products obtained with 1,3,5-trimethoxybenzene contained high levels of acid, probably sulfonated reagent, and were thus unfavorable for analysis by GC–MS. Tests with 2,3,4,5,6-pentafluorobenzyl bromide revealed that only fragments with *m*/*z* 181 and 161 were visible in the positive EI–MS spectrum. These ions had both lost the nitrogen and were therefore not suitable for distinguishing [^14^N] and [^15^N]-nitrate. This is in accordance with the literature, where 2,3,4,5,6-pentafluorobenzyl derivatives are usually analyzed by MS after electron capture negative ion chemical ionization [93,96,97,98]. In addition, 2,3,4,5,6-pentafluorobenzyl bromide reacts also with many other compounds typically present in plants, for instance phenolics [99] and fatty acids [98]. In contrast, mesitylene gave very promising results: a peak at *m*/*z* 148 was obtained for [^14^N]-nitrate (Figure 1B). The small peak at *m*/*z* 149 with a height of approximately 10% of the *m*/*z* 148 peak originates from the natural abundance of the ^13^C and ^15^N isotopes. The observed fragmentation pattern indicated that the ionized molecule with an *m*/*z* of 165 reacted under loss of a hydroxyl radical to the base ion *m*/*z* 148, which reacted further under loss of carbon monoxide to *m*/*z* 120. The latter ion decayed under loss of hydrogen cyanide to *m*/*z* 93. This is in agreement with the fragmentation pattern observed for [^15^N]-nitrate, where peaks of *m*/*z* 149 and 121, one mass unit higher than the [^14^N]-containing ions, were observed. In addition, also for [^15^N]-nitrate *m*/*z* 93 was observed, confirming that this ion does not contain nitrogen.

For benzene, the molecular peak M ^+^ of *m*/*z* 123 obtained by derivatization of [^14^N]-nitrate and of *m*/*z* 124 for [^15^N]-nitrate dominated the spectra (Figure 2). However, also [MH] ^+^ showed a significant peak and thus the molecular peak was not suitable for determination of the [^15^N]/[^14^N]-nitrate ratio. Apart from the molecular peak only the fragments *m*/*z* 107 (for ^14^N) and *m*/*z* 108 for (^15^N) were found to be suitable for determining the [^15^N]/[^14^N]-nitrate ratio since all other fragments lacked nitrogen. However, this signal is relatively weak. In addition, nitrobenzene also shows peak tailing and the reagent benzene is toxic and carcinogenic [100]. Thus, benzene is clearly less suitable than mesitylene.

Taking these results together, mesitylene might be a useful reagent for simple determination of the [^15^N]/[^14^N]-nitrate ratio in plants. Thus, we decided to focus on optimizing the reaction and analysis conditions for this reagent. 

### 2.1. Method Development

Formation of the nitronium cation, the first step in nitration of mesitylene (Figure 1A), depends on the sulfuric acid concentration. Thus, the reaction was performed in the presence of 65–85% (*w*/*w*) sulfuric acid. In addition, the reaction time was varied between 5 and 20 min. A possible side reaction to nitration of the aromatic ring is oxidation of the methyl residues, which happens particularly at an elevated temperature [101,102]. Thus, the reactions were performed at room temperature. Since the mixture was biphasic, the tubes were shaken vigorously to enhance the reaction. For reliable quantification of the reaction product nitromesitylene, 2,4-dinitrotoluene (DNT) was added to the reaction. DNT is a suitable internal standard because the presence of two nitro groups deactivates the aromatic system and thus DNT does not react with nitrate under the conditions tested here. In addition, DNT was well separated from nitromesitylene by GC–MS and the fragment with *m*/*z* 165 can be used for quantification (Figure 3). However, DNT showed significant peak tailing. Since DNT was only needed during optimization of the method, the drawback of DNT peak tailing was considered as acceptable. In the final method DNT is not added to the samples and thus DNT peak tailing has no impact on the accuracy of [^15^N]/[^14^N]-nitrate ratio measurements. 

A clear increase of the nitromesitylene signal was observed with the concentration of sulfuric acid (Figure 4A). At 65% and 70% no reaction was observed while little product was formed at 75%. For the 10 min and 20 min reactions the maximum was reached at 80% sulfuric acid and no further increase of the signal was observed when 85% sulfuric acid was used. For the 5 min reactions, the maximum was observed only for 85% sulfuric acid although a high yield was also obtained at 80%. These data indicate that a sulfuric acid concentration of at least 80% and a reaction time of 10 to 20 min should be used. The obtained reaction product was quite acidic since a lot of carbon dioxide was evolved when the supernatant was treated with sodium carbonate. A possible explanation might be that some sulfuric acid had dissolved in mesitylene. Experiments showed that this problem can be eliminated by addition of water to the tubes after finishing the reactions. To enhance visibility of the phases a small amount of indigo carmine (approximately 0.01%) was added to the water used for dilution of the reaction mix, which stained the lower water/sulfuric acid phase intensively blue while the upper organic phase remained colorless.

For the former experiments, pure mesitylene was used, which served on one hand as reagent and on the other hand as solvent. Mesitylene has a similar boiling point to the reaction product and thus only small volumes could be injected to prevent significant peak broadening of the nitromesitylene peak. In addition, pipette tips were almost invisible since their refractive index (approximately 1.49) is very similar to that of mesitylene (1.499). These disadvantages raised the question whether the amount of mesitylene may be downsized. Reactions with different volumes indicated that the highest signal was obtained with 10 µL mesitylene (the total volume of the organic phase was kept constant at 200 µL by adding heptane after incubation at room temperature for 20 min). At higher volumes, the signal decreased slightly (Figure 4B). Using 80% or 85% sulfuric acid had no significant effect on the reaction. Thus, 80% sulfuric acid and 10 µL mesitylene with a reaction time of 20 min and subsequent addition of water (containing indigo carmine) and 190 µL heptane for extraction of the reaction product were the most suitable conditions. Addition of heptane had also the advantage that pipette tips were well visible in the mesitylene/heptane mixture.

Reactions with different concentrations of nitrate showed perfect linearity for [^14^N]-nitrate and [^15^N]-nitrate and, importantly, an identical detector response (Figure 4C). Also, calibration curves for analysis of the level of [^15^N]-nitrate were linear when mol% [^15^N]-nitrate was plotted on the x-axis and the sum of the intensities of *m*/*z* 149 + 150 divided by the sum of the intensities of *m*/*z* 148 + 149 + 150 plotted on the y-axis (Figure 4D). The ion with *m*/*z* 150 had to be included to correct for nitrate that had reacted with mesitylene containing a ^13^C atom. Without considering ion *m*/*z* 150, a polynomial function of second order rather than a linear function was obtained as a calibration curve.

To investigate the stability of the reaction product a sample was derivatized and analyzed immediately as well as on the following four days. The calculated ratio of [^15^N]-nitrate to [^14^N]-nitrate given in mol% [^15^N]-nitrate (Figure 4E) as well as the ratio of the peak areas of the reaction product nitromesitylene to the internal standard DNT were stable over the entire time tested. This confirms that the reaction product is stable for at least 80 h. 

Initial analyses performed in the splitless injection mode showed relatively broad nitromesitylene peaks with a long tail (Figure 4F). Overloading of the column could be excluded as a reason for peak asymmetry since injection of small volumes gave the same result. However, analyses performed in the split mode showed symmetric peaks even at low split ratios (Figure 4F). Thus, we decided to use an injection volume of 1 µL and a split ratio of 1. Under these conditions sharp symmetric peaks with an approximately equal peak area compared to splitless injection were obtained. 

Separation could be achieved within 6 min using a VF-ms column, making the analysis quick. Chromatograms of standards and leaf samples looked almost identical and showed no peaks interfering with the nitromesitylene peak, confirming that the method is highly specific and suitable for analysis of leaf extracts (Figure 5).

Spiking experiments with plant extracts showed a matrix effect, particularly when samples with high levels of phenolic compounds were measured (Figure 6A). In the presence of 0.57 mM phenolic compounds (calculated as ferulic acid equivalents in the Folin–Ciocalteu assay) a reduction of the signal by approximately one third was observed. In the presence of high levels of phenolic compounds (3.51 mM ferulic acid equivalents) only approximately 20% of the signal intensity of the control was observed. Analysis of [^15^N]-nitrate and [^14^N]-nitrate showed that presence even of high phenolic levels had no effect on the mole fraction (Figure 6B). The matrix effect may be caused by reaction of nitrate with phenolic compounds in the presence of sulfuric acid. Phenolic compounds can be efficiently removed by treatment with activated charcoal [103,104]. The sample containing a high level of phenolic compounds was treated with two types of activated charcoal and with synthetic graphite. Quantification of the phenolic compounds after treatment with the adsorbents showed that both types of activated charcoal removed phenolics efficiently (>0.02 mM residue) while graphite removed only approximately half of the phenolics (1.91 mM after treatment). However, analysis showed that activated charcoal purchased from Merck reduced the matrix effect significantly while activated charcoal from Alfa Aeser and synthetic graphite had only a small effect (Figure 6A). This indicates that also other compounds apart from the phenolics assayed with the Folin–Ciocalteu assay might be responsible for quenching of the signal. Treatment with the adsorbents resulted in significantly lower levels for [^15^N]-nitrate (Figure 6B). Thus, such a treatment is unsuitable for analysis. Since the matrix lowered the signal but did not change the measured [^15^N]-nitrate/[^14^N]-nitrate we decided to use the plant extracts directly. 

The results for [^15^N]-nitrate presented above are the mole fraction in mol%. However, the mole fraction (*x* in mol%) and the ration (*r* in mol/mol) are connected via the formula [105]:(1)$$x=100*\frac{r}{1+r}$$

Thus, if desired, the mole fraction can be converted to the ratio of [^15^N]-nitrate to [^14^N]-nitrate by using the formula:(2)$$r=\;\frac{x}{100-x}$$

### 2.2. Method Validation

The limit of detection (LOD) and limit of quantification (LOQ) for [^15^N]-nitrate in a mixture with [^14^N]-nitrate were 0.65 mol% and 2.2 mol%, respectively. The sample used for determining the LOD and LOQ contained 0.2 mM nitrate since concentrations of 0.1 to 0.6 mM are typically seen for leaf extracts of *Crassocephalum crepidioides* (Appendix A).

For assessing repeatability of the method two samples, one containing a medium and one a high level of phenolic compounds, were analyzed on five consecutive days with five replicates on each day. The calibration functions showed excellent linearity with Pearson correlation coefficients exceeding 0.999 on each day and excellent repeatability with relative standard deviations of 1.8% and 2.2% for the slope and interception, respectively (Table 1). Also, the results obtained for the sample containing a medium level of phenolics were highly repeatable with an intraday SD in the range of 0.25–0.81 mol% and an interday repeatability of 0.87 mol% (Table 2). The results for the sample with a high content of phenolics showed a slightly increased standard deviation with 0.51–2.27% for the intraday SD and 2.18 mol% for the interday SD. This can be explained by the lower signal and thus decreased signal to noise ratio. 

### 2.3. Application: Kinetics of [^14^N]-Nitrate/[^15^N]-Nitrate Replacement in Leaves of Crassocephalum crepidioides

To assess nitrate uptake of *Crassocephalum crepidioides*, an African orphan crop [4,106], plants were grown hydroponically in modified Hoagland medium containing [^15^N]-nitrate as the sole nitrogen source. Since a closed system was used, water was regularly added to maintain a constant volume. In addition, samples were regularly analyzed for minerals and the medium was supplemented if required (Figure 7A). Nitrate and phosphate were the only minerals that had to be supplemented during the course of the experiment. After 29 days, the [^15^N]-nitrate containing medium was replaced by [^14^N]-nitrate medium and from this moment on leaf samples were taken for analysis of foliar [^15^N]-nitrate and [^14^N]-nitrate levels. Initially, a rapid decrease of [^15^N]-nitrate was observed while at later time points a lower decline was observed. Regression analysis showed that [^15^N]-nitrate decreased approximately exponentially with a biological half-life of 4.5 day (Figure 7B). 

## 3. Discussion

For assessing nitrogen metabolism and re-localization of nitrogen-containing metabolites, the heavier nitrogen isotope, ^15^N, has been frequently used [68,72,73,74,75,76,77,78]. In plants it is usually applied in form of nitrate. Nitrate is assimilated mainly in leaves, where it is reduced via nitrite as the intermediate to ammonium, which is rapidly incorporated into amino acids [23,107]. Plants can accumulate high levels of nitrate, which can be critical for re-localization studies since it may take some time until the newly fed labeled nitrate has replaced the previously present unlabeled nitrate. Similarly, in pulse–chase experiments it is important to know to which extent the tracer nitrate has replaced the previously present species since that limits the extent of labeling in downstream metabolites. Thus, for appropriate planning of feeding and pulse–chase experiments knowledge of the kinetics of nitrate uptake and replacement of [^15^N]-nitrate by [^14^N]-nitrate (or vice versa) in leaves is very helpful.

The method for assessing the [^15^N]-nitrate/[^14^N]-nitrate in plant material described here is simple, rapid and does not require special equipment apart from a conventional GC–MS system with EI ionization. Nitrate is extracted with hot water and derivatized at room temperature with mesitylene in the presence of sulfuric acid to nitromesitylene. The reaction product is highly stable and the derivatized samples can be stored at room temperature for at least 80 h without any impact on the performance. The calibration shows perfect linearity and the method is highly reproducible with intraday and interday SDs of less than 0.9 mol% for samples with moderate phenolic contents and less than 2.3 mol% for samples with high phenolic contents. Ideally, the nitrate concentration in the plant extract should be higher than 0.1 mM. At lower concentrations contamination with ubiquitously present nitrate might become critical. The limit of detection and quantification is 0.65 mol% and 2.2 mol% for [^15^N]-nitrate, respectively. The reason for this relatively high detection limit is the natural abundance of ^13^C of approximately 1.1%, which causes even in the absence of [^15^N]-nitrate a signal at *m*/*z* 149 of approximately 10% of that at *m*/*z* 148, the ion obtained for derivatized [^14^N]-nitrate. Opportunities to reduce the detection limit include using a high resolution mass spectrometer that can distinguish the mass defect of [^15^N] and [^13^C] or analyzing derivatized samples by gas chromatography coupled to combustion and reduction units and an isotope ratio mass spectrometer. However, for typical applications using isotope-enriched [^15^N]-nitrate a normal GC–EI–MS system as used here is sufficient.

While the plant matrix had no effect on the results for the [^15^N]-nitrate mole fraction, the signal intensity was significantly reduced, particularly in plant extracts containing high contents of phenolics. Experiments with activated charcoal showed that the matrix effect could be reduced. However, treatment with charcoal had a slight but significant impact on the results for the [^15^N]-nitrate mole fraction. Thus, such a treatment was not useful. Interestingly, while charcoal treatment removed the phenolics as assessed by the Folin–Ciocalteu assay completely, the matrix effect was not completely eliminated. That indicates that the nitronium ion may also react with other compounds present in the plant extract apart from phenolics. Possible candidates are reducing compounds like ascorbic acid, glutathione, or cysteine [108,109]. However, since the measured [^15^N]-nitrate mole fraction was not affected by the plant matrix this was not studied in more detail.

In summary, the presented method allows simple and reproducible determination of the foliar [^15^N]-nitrate/[^14^N]-nitrate ratio even from small samples. The method can be helpful for establishing and evaluating feeding and pulse–chase experiments.

## 4. Materials and Methods

### 4.1. Reagents

Potassium nitrate (>99%) was purchased from Duchefa (Haarlem, the Netherlands). [^15^N]-Potassium nitrate (>98 isotope purity), mesitylene, heptane, indigo carmine, 2,4-dinitrotoluene, 1,3,5-trimethoxybenzene, toluene, 2,3,4,5,6-pentafluorobenzyl bromide, ethyl acetate, acetone, sodium tetraborate, and benzyldimethyltetradecylammonium chloride were purchased from Sigma (St. Louis, MO, USA). Sulfuric acid 96% and benzene were purchased from Carl Roth (Karlsruhe, Germany). 

### 4.2. Plant Material and Growth Conditions

*Crassocephalum crepidioides* accession Ilé-Ifè [106] was used in this study. Seedlings were transplanted into hydroponic medium after 4 weeks of pre-cultivation on soil. Each hydroponic tank harbored 6 plants and contained 4 L of modified Hoagland medium with [^15^N]-nitrate as the sole nitrogen source. Since a relatively low nitrate concentration was required the calcium nitrate present in Hoagland medium was replaced by calcium chloride (for medium composition and preparation see Appendix A). Thus, the medium contained a relatively high but constant chloride concentration. Constant aeration of the medium was provided by air stones connected to an air pump. Plants were cultivated at 21 °C with a 16 h light/8 h dark cycle and a light intensity of 100 µmol∙s^−1^∙m^−2^. After 29 days in the initial medium plants were switched to medium containing [^14^N]-nitrate. Roots were rinsed with deionized water before transfer to the other medium. Sampling of leaf discs was performed over a 19-day period.

Plants used for preparation of nitrate-free leaf extracts used for investigating the matrix effect (see point 4.4.6.) were cultivated on low nutrient soil SP ED63 P (Patzer GmbH and Co. KG, Sinntal-Altengronau, Germany; initial nitrate content: 555 mg/kg dry weight) for 12 weeks. The plants showed nitrogen deficiency symptoms three weeks prior to harvest and absence of nitrate in the leaf extracts was confirmed by ion-pair chromatography and by GC–MS.

### 4.3. Sample Preparation and GC–MS

Leaf discs with 8 mm diameter (approximately 10 mg) were cut out using a hollow punch, placed into 2 mL safe lock tubes and stored at –20 °C until analysis. Distilled water (150 µL) was added and the tube incubated in a Thermomix shaker (Eppendorf, Hamburg, Germany) set to 1400 rpm and 95 °C for 20 min. After centrifugation at 10,000 rpm for 2 min the nitrate concentration was quantified by ion-pair chromatography (see Section 4.7). and the remaining extract diluted to a final nitrate concentration of 0.2–0.4 mM if required. A 2 mL safe lock tube was charged with 100 µL extract, 600 µL 80% sulfuric acid, and 10 µL mesitylene. The tubes were placed in a Thermomix shaker set to 1400 rpm at room temperature for 20 min. Water (500 µL) containing 0.01% indigo carmine and 190 µL heptane were added and the tubes shaken vigorously prior centrifugation at 10,000 rpm for 30 s. Approximately 150 µL of the supernatant was transferred into a 1.5 mL reaction tube containing approximately 5 mg sodium carbonate and mixed vigorously. After centrifugation at 10,000 rpm for 30 s, 80 µL of the clear supernatant was transferred into an autosampler tube. 

A Varian 431-GC gas chromatograph equipped with a CP-8400 autosampler and connected to a 210-MS mass spectrometer (Palo Alto, CA, USA) was used. For analysis, a VF-5ms 30 m × 0.25 mm × 0.25 µm capillary column was used. Helium was used as carrier gas at a flow rate of 1 mL/min and a split ration of 10. The injector was set to 230 °C. Injection (1 µL sample) was performed at a split ratio of 1. The temperature program started with an isothermal step at 120 °C for 1 min. Then the temperature was raised linearly within 4 min to 200 °C at a rate of 20 °C/min. Finally, the temperature was raised within 1 min to 260 °C at a rate of 60 °C prior returning to the initial conditions. The transfer line was operated at 200 °C, the ion trap at 160 °C, and the manifold at 40 °C. MS spectra were recorded from 3.6 to 4.8 min and from *m*/*z* 50 to 200. For establishing the calibration curve the sum of the intensities of the ions with *m*/*z* 149 and 150 was divided by the sum of the ions with *m*/*z* 148 + 149 + 150 and plotted on the y-axis while mol% of [^15^N]-nitrate was plotted on the x-axis. A detailed step-by-step protocol is included in Appendix B.

### 4.4. Method Development

#### 4.4.1. Initial Testing of Different Derivatization Methods

Derivatization with mesitylene was performed according to Dunphy et al. [90] except that no internal standard was added. Derivatization with 1,3,5-trimetrhoxybenzen was performed as described by Gutzki et al. [92] except that the reaction product was extracted with toluene rather than benzene. For derivatization with 2,3,4,5,6-pentafluorobenzyl bromide the protocols of Tsikas et al. [94] and of Kage et al. [93] were applied. Derivatization with benzene was adapted from [89]. In brief, 100 µL of sample was mixed with 600 µL of 85% sulfuric acid and 200 µL of benzene in 2 mL safe lock tubes. The tubes were shaken at room temperature in an Eppendorf Thermomixer (Hamburg, Germany) set to 1400 rpm for 15 min. The upper layer was transferred to a tube containing approximately 5 mg sodium carbonate, shaken vigorously and centrifuged at 10,000 × *g* for 30 s. The supernatant was directly analyzed. The reaction product with benzene was analyzed by GC–MS as stated above except that the temperature program was modified: isothermal at 100 °C for 1 min, then a liner increase to 180 °C within 4 min, and finally a linear increase to 260 °C within 2 min prior returning to the initial conditions. The other derivatization products were analyzed using the settings mentioned in Section 4.4.2.

#### 4.4.2. Optimization of Reaction Time and Sulfuric Acid Concentration

Sample (100 µL; 1 mM [^14^N]-potassium nitrate) was mixed with 600 µL sulfuric acid of the indicated concentration and 300 µL mesitylene containing 0.5 mM 2,4-dinitrotoluene as the internal standard. The mixture was derivatized and processed as stated above except that a reaction time of 5, 10, or 20 min was used. The clear supernatant was analyzed by GC–MS as described above except that the following GC settings were used: injector temperature 250 °C; thermal program: isothermal at 120 °C for 1 min, then a linear increase to 230 °C within 5.5 min, and finally a linear increase to 290 °C within 1.5 min prior returning to the initial conditions. MS spectra were recorded from 3.6 to 4.8 min and from *m*/*z* 50 to 200.

#### 4.4.3. Optimization of Mesitylene Amount and Sulfuric Acid Concentration

Sample (100 µL; 0.5 mM [^14^N]-potassium nitrate) was mixed with 600 µL of 80% or 85% sulfuric acid and 5, 10, 20, 50, 100, or 150 µL mesitylene was added. The mixture was shaken at 1400 rpm at room temperature for 20 min. The reaction was stopped by addition of 500 µL of water containing 0.01% indigo carmine. Subsequently, 50 µL of 0.5 mM 2,4-dinitrotoluene dissolved in mesitylene was added as the internal standard and heptane was added to complete the organic phase to 200 µL. The supernatants were processed and analyzed as described in Section 4.4.2. 

#### 4.4.4. Test for Linearity

Potassium nitrate solutions (100 µL of [^14^N]-potassium nitrate or 100 µL of [^15^N]-potassium nitrate) of different concentrations (0, 0.005, 0.01, 0.02, 0.05, 0.1, 0.2, 0.5, and 1 mM) were analyzed as described in Section 4.4.3. 

#### 4.4.5. Stability of the Reaction Products

A sample was derivatized with 20 µL of mesitylene as described in Section 4.4.3. The derivatized solution was measured on five consecutive days with five replicates on each day. The derivatized sample was stored at room temperature.

#### 4.4.6. Matrix Effect

Nitrate-free plant extracts obtained from *C. crepidioides* leaves were spiked with a mixture of [^14^N] and [^15^N]-potassium nitrate to a total concentration of 0.4 mM nitrate. The phenolic contents were assayed as described in Section 4.8. [^15^N] and [^14^N]-nitrate were measured as described in Section 4.4.3.

### 4.5. Method Validation

A sample containing 0.4 mM [^14^N]-nitrate was measured 9 times and the average and SD were calculated. The LOD and LOQ were calculated by dividing the 3-fold and 10-fold SD by the slope of the calibration curve, respectively. 

For testing the repeatability, samples with a medium and high content of phenolics were spiked with a mixture of [^14^N] and [^15^N]-potassium nitrate to a total concentration of 0.4 mM nitrate. The samples were analyzed on five consecutive days with five replicates on each day. Every day a calibration curve (see Appendix B for details) was established.

### 4.6. Quantification of Nitrate in Leaf Extracts by Ion-Pair Chromatography

Quantification of nitrate by ion-pair chromatography was adapted from Chou et al. [60]. A Shimadzu 10A high performance liquid chromatography system (Shimadzu, Kyoto, Japan) was used. The HPLC system consisted of a SCL-10A system controller, a FCV-10AL valve for eluent selection, a LC-10AT pump equipped with DGU-14A inline degasser, a SIL-10A autosampler, a CTO-10ASvp column oven set to 25 °C, a Nucleodur 100-5 C18ec 125 × 4.6 mm HPLC column (Machery-Nagel, Düren, Germany), and a SPD-10A UV detector, which was operated at 213 nm. The eluent consisted of 10 mM 1-octylamine set with phosphoric acid to pH 7.0 in 20% ACN (acetonitrile). For elution, a flow rate of 1 mL/ min was set and the injection volume was 25 µL. Samples were diluted with eluent 1:10 prior to injection. Standards were dissolved in eluent and contained nitrate in the range of 0–10 mg/L. Chromatograms were evaluated with the Clarity software package (version 5.0.4.158, DataApex, Prague, Czech Republic).

### 4.7. Analysis of Cations and Anions in Hydroponic Media by Ion Chromatography

For analysis of cations the samples were diluted 1:50 by transferring 1 mL of sample into a 50 mL volumetric flask and adding deionized water to the mark. An aliquot was transferred into a polypropylene autosampler vial and 100 µL was injected into the ion chromatograph. The ion chromatograph consisted of a LC-10ADvp pump, a FCV-10ALvp valve for eluent selection, a SIL-10A autosampler, a CTO-10ASvp column (Shimadzu), a Nucleodur 100-5 C18ec 4 × 3 mm pre-column, an IC-Pak Cation M/D 3.9 × 150 mm column (Waters, Milford, MA, USA), and a 430 conductivity detector (Waters). Chromatograms were evaluated with the Clarity software package. The column oven was maintained at 25 °C. The eluent consisted of 3 mM nitric acid and 0.1 mM EDTA (ethylenediaminetetraacetic acid) and was delivered at a flow rate of 1 mL/min. Standards contained sodium (0–0.05 mM), ammonium (0–0.05 mM), potassium (0–0.25 mM), magnesium (0–0.25 mM), and calcium (0–0.5 mM).

Anions were analyzed by diluting the samples 1:5 by transferring 1 mL of sample into a 5 mL volumetric flask and adding deionized water to the mark. An aliquot was transferred into a polypropylene autosampler vial and 100 µL was injected into the ion chromatograph described above except that an IC-PAK Anion HC 150 × 4.6 mm column (Waters) and a sodium gluconate/borate buffer (sodium gluconate 320 mg/L, boric acid 360 mg/L, disodium tetraborate 264 mg/L, glycerol 4 g/L, acetonitrile 12%, and 1-butanol 2%) delivered at a flow rate of 1.6 mL/min were used. The standards contained chloride (0–6.42 mM), nitrate (0–1.7 mM), phosphate (0–0.4 mM), and sulfate (0–1.81 mM).

### 4.8. Determination of Total Phenolics in Leaf Extracts

The method was adapted from Veliogli et al. [110]. In brief, suitable volumes of sample or standard (0–50 µL of 0.1 mM ferulic acid) were transferred into 1 mL plastic cuvettes and water was added to a total volume of 700 µL. Subsequently, 50 µL of Folin–Ciocalteu reagent was added. After mixing and incubating at room temperature for 5 min, 250 µL of 2 M sodium carbonate was added and mixed well. The absorbance was measured at 725 nm after incubation for 1 h at room temperature. The results were expressed in mM ferulic acid equivalents. 

## Figures and Tables

**Figure 1 molecules-24-01531-f001:**
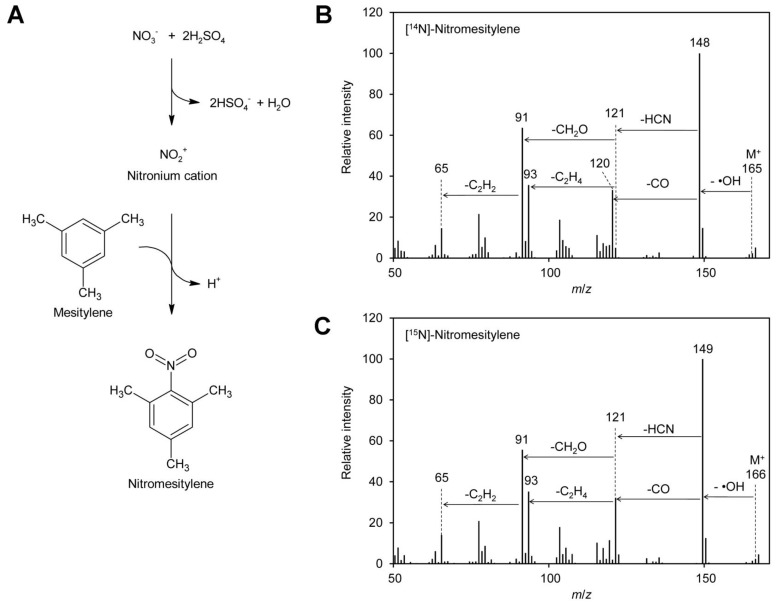
Formation of nitromesitylene and MS spectra. (**A**) Nitrate reacts with sulfuric acid to the nitronium cation, which reacts subsequently with mesitylene to form nitromesitylene. (**B**) EI–MS spectrum of [^14^N]-nitromesitylene. Neutral losses of a hydroxyl radical (·OH), carbon monoxide (CO), and hydrogen cyanide (HCN) are indicated. (**C**) EI–MS spectrum of [^15^N]-nitromesitylene. Original data are presented in Appendix A.

**Figure 2 molecules-24-01531-f002:**
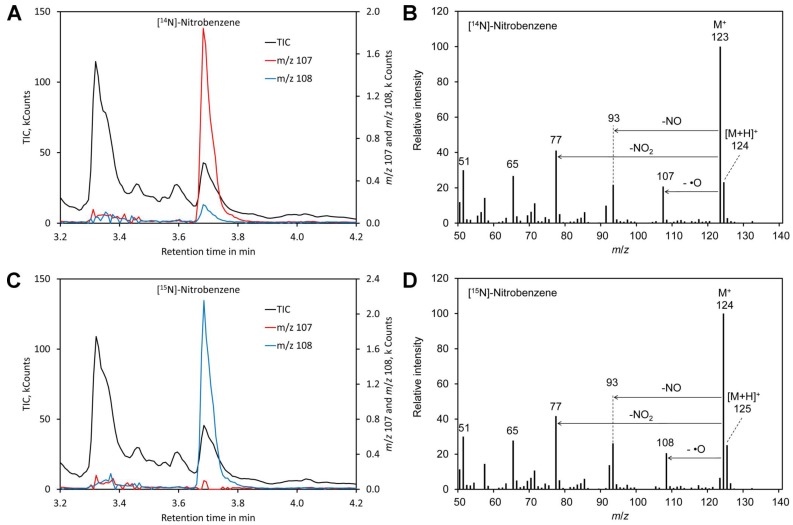
Derivatization with benzene. (**A**) Chromatogram obtained by derivatization of [^14^N]-nitrate. (**B**) Mass spectrum of the derivatization product [^14^N]-nitrobenzene. (**C**) Chromatogram obtained by derivatization of [^14^N]-nitrate. (**D**) Mass spectrum of the derivatization product [^14^N]-nitrobenzene. Original data are included in Appendix A.

**Figure 3 molecules-24-01531-f003:**
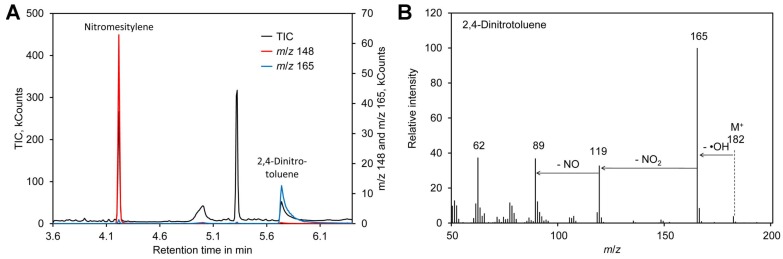
2,4-Dinitrotoluene as internal standard. (**A**) Chromatogram obtained by derivatization of [^14^N]-nitrate with nitromesitylene containing 0.5 mM 2,4-dinitrotoluene. (**B**) Mass spectrum of 2,4-dinitrotoluene. Original data are included in Appendix A.

**Figure 4 molecules-24-01531-f004:**
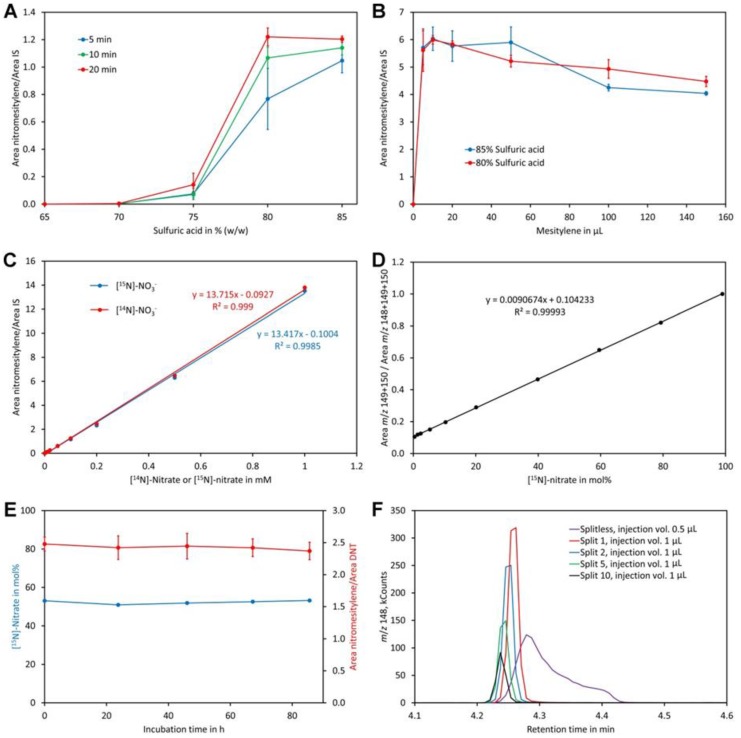
Optimization of the method. (**A**) Reaction time and sulfuric acid concentration. The dots and bars are the average and SD of three replicates, respectively. (**B**) Amount of mesitylene and sulfuric acid concentration. The dots and bars are the average and SD of three replicates, respectively. (**C**) Linearity in the range of 0 to 1 mM [^14^N]-nitrate and [^15^N]-nitrate. (**D**) Calibration curve for determination of the [^15^N]-nitrate level using standards containing 0.2 mM nitrate. (**E**) Stability of the reaction product. The derivatized sample was injected after incubation for the indicated time. The blue dots represent the measured content of [^15^N]-nitrate in mol% while the red dots represent the ratio of the area of nitromesitylene to the area of the internal standard 2,4-dinitrotoluene (DNT). The dots and bars are the average and SD of five measurements, respectively. (**F**) Testing different injection modes. The same sample was injected splitless or with a split ratio of 1, 2, 5, or 10. The injection volume was 1 µL except for splitless injection, where 0.5 µL was injected. Original data are presented in Appendix A.

**Figure 5 molecules-24-01531-f005:**
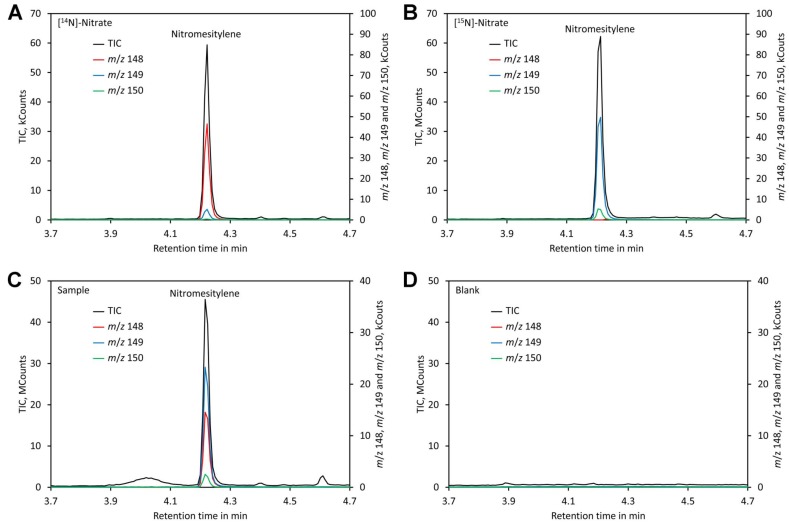
Chromatograms. (**A**) Chromatogram of a standard containing 0.2 mM [^14^N]-nitrate. (**B**) Chromatogram of a standard containing 0.2 mM [^15^N]-nitrate. (**C**) Chromatogram of a plant sample grown on medium containing [^15^N]-nitrate and shifted to medium containing [^14^N]-nitrate 3 days prior analysis. (**D**) Chromatogram of a blank. Original data are presented in Appendix A.

**Figure 6 molecules-24-01531-f006:**
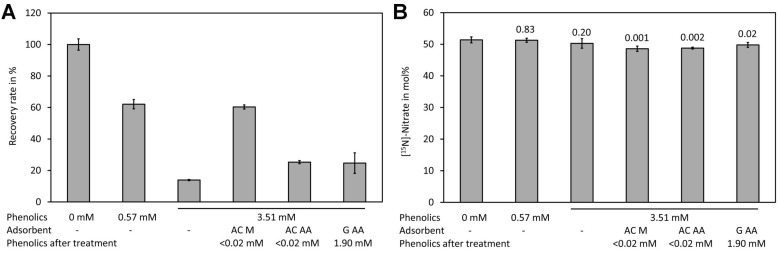
Matrix effect of plant leaf extracts. (**A**) Samples containing different levels of phenolic compounds (measured in mM ferulic acid equivalents) were spiked with a mixture containing [^14^N]-nitrate and [^15^N]-nitrate to a final concentration of 0.4 mM total nitrate and treated with the indicated absorbents. Subsequently, the residual phenolics and the recovery rate of total nitrate were assayed using DNT as the internal standard. (**B**) The [^15^N]-nitrate levels of the same samples shown in (**A**). Original data are presented in Appendix A.

**Figure 7 molecules-24-01531-f007:**
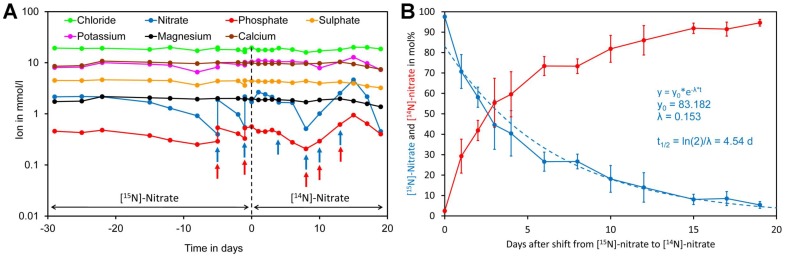
Kinetics of replacement of [^15^N]-nitrate by [^14^N]-nitrate in leaves of *Crassocephalum crepidioides*. (**A**) Plants were grown hydroponically in liquid medium containing [^15^N]-nitrate as the sole nitrogen source for 29 days. Subsequently, the medium was replaced by medium containing [^14^N]-nitrate as the sole nitrogen source (time 0, indicated by a dotted line) and grown for a further 19 days. Samples of the medium were taken at the indicated time points and analyzed for ions. If required, the medium was supplemented for nitrate by adding [^15^N] or [^14^N]-potassium nitrate (blue arrows) or for phosphate by adding potassium dihydrogen phosphate (red arrows). (**B**) Leaf disc samples were taken at the indicated time points for analysis of leaf [^14^N]-nitrate and [^15^N]-nitrate levels. The points and bars represent the biological average and standard error (SE) of samples taken from four different leaves. Total nitrate concentrations in leaf material and original data are presented in Appendix A.

**Table 1 molecules-24-01531-t001:** Interday repeatability of the calibration curve.

Experiments ^1^	Calibration ^2^		
	a	b	r
Day 1	0.00866	0.10698	0.99915
Day 2	0.00908	0.09956	0.99991
Day 3	0.00895	0.10065	0.99988
Day 4	0.00879	0.10223	0.99980
Day 5	0.00890	0.10155	0.99982
Average	0.00888	0.10220	
SD	0.00016	0.00286	
RSD in%	1.78	2.80	

^1^ Original data are presented in Appendix A; ^2^ Linear regression function of the type y = a × x + b; a, slope; b, interception; r, Pearson correlation coefficient.

**Table 2 molecules-24-01531-t002:** Intraday and interday repeatability.

Experiment ^1^	Repeats	Reproducibility		
		Average	SD	RSD
		mol%	mol%	%
0.57 mM FA				
Day 1	5	53.10	0.57	1.07
Day 2	5	50.94	0.46	0.91
Day 3	5	51.99	0.25	0.48
Day 4	5	51.84	0.25	0.48
Day 5	5	51.33	0.81	1.58
Interday	25	51.84	0.88	1.70
3.51 mM FA				
Day 1	5	52.37	0.51	0.98
Day 2	5	48.75	2.27	4.65
Day 3	5	48.49	2.12	4.37
Day 4	5	49.86	0.99	1.99
Day 5	5	50.49	0.99	1.97
Interday	25	49.99	2.00	4.00

^1^ Original data are presented in Appendix A.

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
