# Peer review of "Determination of the [15N]-Nitrate/[14N]-Nitrate Ratio in Plant Feeding Studies by GC–MS"

_molecules, 2019, doi:10.3390/molecules24081531_

Round 1

Reviewer 1 Report

Reviewer report on manuscript Molecules-483492

The submitted approach aimed on the development of a GC-MS method for the determination of [15N]-nitrate/[14N]-nitrate ratio in plant feeding studies. 

I read throughout the manuscript. It is well-structured, referenced and figured. I agree with the authors regarding to the novelty of their method. However, there are some minor comments that need to be addressed prior to its final acceptance.

Comments

1)   Line 191: The studied concentration range of sulfuric acid must be mentioned.

2)   Section 2.1: According to the manuscript the DNT (2,4-dinitrotoluene) has been utilized as internal standard although that a considerable peak tailing was observed. To my opinion this would may lead on problematic integration of the particular peak and therefore in the final results. Did the authors try to use another compound (e.g nitro-toluene)?

3)   Line 240: a more appropriate term for “bent curve” should be used.

4)   Table 1 caption: correct to “calibration curve”

Author Response

Response to Reviewer 1 Comments

Reviewer 1

Reviewer report on manuscript Molecules-483492

The submitted approach aimed on the development of a GC-MS method for the determination of [15N]-nitrate/[14N]-nitrate ratio in plant feeding studies.

I read throughout the manuscript. It is well-structured, referenced and figured. I agree with the authors regarding to the novelty of their method. However, there are some minor comments that need to be addressed prior to its final acceptance.

 Response:

Many thanks for the positive evaluation of our manuscript and for the helpful comments, which we have addressed as outlined below.

Comment 1:

Line 191: The studied concentration range of sulfuric acid must be mentioned.

Response 1:

We agree that the sulfuric acid concentration should be mentioned. We have adapted the sentence accordingly.

Comment 2:

Section 2.1: According to the manuscript the DNT (2,4-dinitrotoluene) has been utilized as internal standard although that a considerable peak tailing was observed. To my opinion this would may lead on problematic integration of the particular peak and therefore in the final results. Did the authors try to use another compound (e.g nitro-toluene)? 

Response 1:

It is correct that DNT shows strong chemical tailing, which makes integration of the peak area less accurate. However, DNT was only required during optimisation of the method but it is not included in the final protocol. Thus, DNT tailing has no impact on the accuracy of [14N]/[15N] ratio determinations. Since DNT was only needed for initial experiments for optimisation we considered the small error caused by tailing of the DNT peak acceptable. We have adapted the text to explain in more detail that DNT was only needed during optimisation and thus tailing of the DNT peak has no impact on the accuracy of the final method.

Comment 3:

Line 240: a more appropriate term for “bent curve” should be used.

Response 3:

We agree that “bent curve” is not appropriate. We have replaced it by “polynomial function of second order”.

Comment 4:

Table 1 caption: correct to “calibration curve”.   

Response 4:

The caption was corrected as suggested.

Reviewer 2 Report

Comments and Suggestions for Authors

The paper entitled “Determination of the [15N]-Nitrate/[14N]-Nitrate Ratio in Plant Feeding Studies by GC-MS” aims to tested whether these methods can be applied for determining the [15N]-nitrate/[14N]-nitrate ratio in plant material using GC-MS with electron impact ionization.. It is an interesting paper, well prepared, covering a lot of relevant information.

I have no specific scientific concern but it is mandatory to correct the manuscript in some points:

The introduction is a little too long.

#line 42:

Is ‘Plants grown in nitrogen-rich substrates may contain more than 5% nitrogen in dry weight while plants grown under nitrogen-limiting conditions contain still approximately 0.5-1% nitrogen in dry weight [10,11].

Should be ‘Plants grown in nitrogen-rich substrates may contain more than 5% nitrogen in dry weight while plants grown under nitrogen-limiting conditions contain still approximately 0.5-1.0% nitrogen in dry weight [10,11].

Should be: ”Reactions with different volumes indicated that the highest signal was obtained with 10 μL mesitylene (the total volume of the organic phase was kept constant at 200 μL by adding heptane after incubation at room temperature for 20 min). At higher volumes, the signal decreased slightly (Figure 4B). Using 80% or 85% sulfuric acid had no significant effect on the reaction. Thus, 80% sulfuric acid and 10 μL mesitylene with a reaction time of 20 min and subsequent addition of water (containing indigo carmine) and 190 μL heptane for extraction of the reaction product were the most suitable conditions.

#line 198

Is: “In addition, DNT is well separated from nitromesitylene by GC-MS and the fragment with m/z 165 can be used for quantification (Figure 3).”

Should be: “In addition, DNT is well separated from nitromesitylene by GC-MS and the fragment with m/z 165 can be used for quantification (Figure 3).”

#line 265

Is: “Thus, we decided to use an injection volume of 1 μl and a split ratio of 1.

Should be: “Thus, we decided to use an injection volume of 1 μL and a split ratio of 1.

#line 226

Is: ”Reactions with different volumes indicated that the highest signal was obtained with 10 μl mesitylene (the total volume of the organic phase was kept constant at 200 μl by adding heptane after incubation at room temperature for 20 min). At higher volumes, the signal decreased slightly (Figure 4B). Using 80% or 85% sulfuric acid had no significant effect on the reaction. Thus, 80% sulfuric acid and 10 μl mesitylene with a reaction time of 20 min and subsequent addition of water (containing indigo carmine) and 190 μl heptane for extraction of the reaction product were the most suitable conditions.

#line 416

Is: Each hydroponic tank harbored 6 plants and contained 4 l of modified Hoagland medium with [15N]-nitrate as sole nitrogen source.

Should be: Each hydroponic tank harbored 6 plants and contained 4 L of modified Hoagland medium with [15N]-nitrate as sole nitrogen source.

#line 434

Is: “Distilled water (150 μl) was added and the tubed incubated in a Thermomix shaker (Eppendorf, Hamburg, Germany) set to 1400 rpm and 95°C for 20 min. After centrifugation at 10000 rpm for 2 min the nitrate concentration was quantified by ion-pair chromatography (see 4.7.) and the remaining extract diluted to a final nitrate concentration of 0.2-0.4 mM if required. A 2 ml safe lock tube was charged with 100 μl extract, 600 μl 80% sulfuric acid and 10 μl mesitylene.”

Should be: “Distilled water (150 μL) was added and the tubed incubated in a Thermomix shaker (Eppendorf, Hamburg, Germany) set to 1400 rpm and 95°C for 20 min. After centrifugation at 10000 rpm for 2 min the nitrate concentration was quantified by ion-pair chromatography (see 4.7.) and the remaining extract diluted to a final nitrate concentration of 0.2-0.4 mM if required. A 2 mL safe lock tube was charged with 100 μL extract, 600 μL 80% sulfuric acid and 10 μL mesitylene.”

#line 440

Is: “Water (500 μl) containing 0.01% indigo carmine and 190 μl heptane were added and the tubes shaken vigorously prior centrifugation at 10000 rpm for 30 s. Approximately 150 μl of the supernatant was transferred into a 1.5 ml reaction tube containing approximately 5 mg sodium carbonate and mixed vigorously. After centrifugation at 10000 rpm for 30 s, 80 μl of the clear supernatant was transferred into an autosampler tube.

Should be: Water (500 μlL containing 0.01% indigo carmine and 190 μL heptane were added and the tubes shaken vigorously prior centrifugation at 10000 rpm for 30 s. Approximately 150 μL of the supernatant was transferred into a 1.5 mL reaction tube containing approximately 5 mg sodium carbonate and mixed vigorously. After centrifugation at 10000 rpm for 30 s, 80 μL of the clear supernatant was transferred into an autosampler tube.

#line 464

Is: “Derivatization with benzene was adapted from [93]. In brief, 100 μl sample were mixed with 600 μl 85% sulfuric acid and 200 μl benzene in 2 ml safe lock tubes. The tubes were shaken at room temperature in a Thermomixer set to 1400 rpm for 15 min

Should be: “Derivatization with benzene was adapted from [93]. In brief, 100 μL sample were mixed with 600 μL 85% sulfuric acid and 200 μL benzene in 2 mL safe lock tubes. The tubes were shaken at room temperature in a Thermomixer set to 1400 rpm for 15 min

#line 475

Is: “Sample (100 μl; 1 mM [14N]-potassium nitrate) was mixed with 600 μl sulfuric acid of the indicated concentration and 300 μl mesitylene containing 0.5 mM 2,4-dinitrotoluene as internal standard.

Should be: “Sample (100 μL; 1 mM [14N]-potassium nitrate) was mixed with 600 μL sulfuric acid of the indicated concentration and 300 μL mesitylene containing 0.5 mM 2,4-dinitrotoluene as internal standard.

#line 485

Is: “Sample (100 μl; 0.5 mM [14N]-potassium nitrate) was mixed with 600 μl sulfuric acid 80% or 85% and 5, 10, 20, 50, 100 or 150 μl mesitylene was added. The mixture was shaken with 1400 rpm at room temperature for 20 min. The reaction was stopped by addition of 500 μl water containing 0.01% indigo carmine. Subsequently, 50 μl 0.5 mM 2,4-dinitrotoluene dissolved in mesitylene was added as internal standard and heptane was added to complete the organic phase to 200 μl.

Should be: “Sample (100 μL; 0.5 mM [14N]-potassium nitrate) was mixed with 600 μL sulfuric acid 80% or 85% and 5, 10, 20, 50, 100 or 150 μL mesitylene was added. The mixture was shaken with 1400 rpm at room temperature for 20 min. The reaction was stopped by addition of 500 μl water containing 0.01% indigo carmine. Subsequently, 50 μL 0.5 mM 2,4-dinitrotoluene dissolved in mesitylene was added as internal standard and heptane was added to complete the organic phase to 200 μL.

#line 493

Is: “Potassium nitrate solutions (100 μl of [14N]-potassium nitrate or 100 μl of [15N]-potassium nitrate) of different concentrations (0, 0.005, 0.01, 0.02, 0.05, 0.1, 0.2, 0.5 and 1 mM) were analyzed as described in 4.4.3.

Should be: “Potassium nitrate solutions (100 μL of [14N]-potassium nitrate or 100 μL of [15N]-potassium nitrate) of different concentrations (0, 0.005, 0.01, 0.02, 0.05, 0.1, 0.2, 0.5 and 1 mM) were analyzed as described in 4.4.3.

#line 498:

Is: “A sample was derivatized with 20 μl mesitylene as described in 4.4.3.

Should be: “A sample was derivatized with 20 μL mesitylene as described in 4.4.3.

Author Response

Response to Reviewer 2 Comments

Reviewer 2

The paper entitled “Determination of the [15N]-Nitrate/[14N]-Nitrate Ratio in Plant Feeding Studies by GC-MS” aims to tested whether these methods can be applied for determining the [15N]-nitrate/[14N]-nitrate ratio in plant material using GC-MS with electron impact ionization.. It is an interesting paper, well prepared, covering a lot of relevant information.

I have no specific scientific concern but it is mandatory to correct the manuscript in some points.

Response:

Many thanks for the positive evaluation of our manuscript and for the helpful comments, which we have addressed as outlined below.

Comment 1:

The introduction is a little too long.

Response 1:

We agree that the introduction is long; particularly the part about isotope discrimination is beyond the focus of this manuscript. We have deleted that part (lines 93-102 of the former manuscript) and restructured the text accordingly.

Comment 2:

#line 42:

Is ‘Plants grown in nitrogen-rich substrates may contain more than 5% nitrogen in dry weight while plants grown under nitrogen-limiting conditions contain still approximately 0.5-1% nitrogen in dry weight [10,11].’

Should be ‘Plants grown in nitrogen-rich substrates may contain more than 5% nitrogen in dry weight while plants grown under nitrogen-limiting conditions contain still approximately 0.5-1.0% nitrogen in dry weight [10,11].’

Response 1:

The sentence was corrected as suggested.

Comment 3:

#line 198

Is: “In addition, DNT is well separated from nitromesitylene by GC-MS and the fragment with m/z 165 can be used for quantification (Figure 3).”

Should be: “In addition, DNT is well separated from nitromesitylene by GC-MS and the fragment with m/z 165 can be used for quantification (Figure 3).”.

Response 3:

The sentence was corrected by replacing “is” by “was”.

Comment 4:

#line 265

Is: “Thus, we decided to use an injection volume of 1 μl and a split ratio of 1.”

Should be: “Thus, we decided to use an injection volume of 1 μL and a split ratio of 1.”

Response 4:

The sentence was corrected as suggested.

Comment 5:

#line 226

Is: ”Reactions with different volumes indicated that the highest signal was obtained with 10 μl mesitylene (the total volume of the organic phase was kept constant at 200 μl by adding heptane after incubation at room temperature for 20 min). At higher volumes, the signal decreased slightly (Figure 4B). Using 80% or 85% sulfuric acid had no significant effect on the reaction. Thus, 80% sulfuric acid and 10 μl mesitylene with a reaction time of 20 min and subsequent addition of water (containing indigo carmine) and 190 μl heptane for extraction of the reaction product were the most suitable conditions.”

 Should be: ”Reactions with different volumes indicated that the highest signal was obtained with 10 μL mesitylene (the total volume of the organic phase was kept constant at 200 μL by adding heptane after incubation at room temperature for 20 min). At higher volumes, the signal decreased slightly (Figure 4B). Using 80% or 85% sulfuric acid had no significant effect on the reaction. Thus, 80% sulfuric acid and 10 μL mesitylene with a reaction time of 20 min and subsequent addition of water (containing indigo carmine) and 190 μL heptane for extraction of the reaction product were the most suitable conditions.”

Response 5:

The paragraph was corrected as suggested.

Comment 6:

#line 416

Is: Each hydroponic tank harbored 6 plants and contained 4 l of modified Hoagland medium with [15N]-nitrate as sole nitrogen source.”

Should be: Each hydroponic tank harbored 6 plants and contained 4 L of modified Hoagland medium with [15N]-nitrate as sole nitrogen source.”

Response 6:

The sentence was corrected as suggested.

Comment 7:

#line 434

Is: “Distilled water (150 μl) was added and the tubed incubated in a Thermomix shaker (Eppendorf, Hamburg, Germany) set to 1400 rpm and 95°C for 20 min. After centrifugation at 10000 rpm for 2 min the nitrate concentration was quantified by ion-pair chromatography (see 4.7.) and the remaining extract diluted to a final nitrate concentration of 0.2-0.4 mM if required. A 2 ml safe lock tube was charged with 100 μl extract, 600 μl 80% sulfuric acid and 10 μl mesitylene.”

Should be: “Distilled water (150 μL) was added and the tubed incubated in a Thermomix shaker (Eppendorf, Hamburg, Germany) set to 1400 rpm and 95°C for 20 min. After centrifugation at 10000 rpm for 2 min the nitrate concentration was quantified by ion-pair chromatography (see 4.7.) and the remaining extract diluted to a final nitrate concentration of 0.2-0.4 mM if required. A 2 mL safe lock tube was charged with 100 μL extract, 600 μL 80% sulfuric acid and 10 μL mesitylene.”

Response 7:

The paragraph was corrected as suggested.

Comment 8:

#line 440

Is: “Water (500 μl) containing 0.01% indigo carmine and 190 μl heptane were added and the tubes shaken vigorously prior centrifugation at 10000 rpm for 30 s. Approximately 150 μl of the supernatant was transferred into a 1.5 ml reaction tube containing approximately 5 mg sodium carbonate and mixed vigorously. After centrifugation at 10000 rpm for 30 s, 80 μl of the clear supernatant was transferred into an autosampler tube.”

Should be: Water (500 μlL containing 0.01% indigo carmine and 190 μL heptane were added and the tubes shaken vigorously prior centrifugation at 10000 rpm for 30 s. Approximately 150 μL of the supernatant was transferred into a 1.5 mL reaction tube containing approximately 5 mg sodium carbonate and mixed vigorously. After centrifugation at 10000 rpm for 30 s, 80 μL of the clear supernatant was transferred into an autosampler tube.”

Response 8:

The paragraph was corrected as suggested.

Comment 9:

#line 464

Is: “Derivatization with benzene was adapted from [93]. In brief, 100 μl sample were mixed with 600 μl 85% sulfuric acid and 200 μl benzene in 2 ml safe lock tubes. The tubes were shaken at room temperature in a Thermomixer set to 1400 rpm for 15 min”

Should be: “Derivatization with benzene was adapted from [93]. In brief, 100 μL sample were mixed with 600 μL 85% sulfuric acid and 200 μL benzene in 2 mL safe lock tubes. The tubes were shaken at room temperature in a Thermomixer set to 1400 rpm for 15 min”

Response 9:

The sentence was corrected as suggested.

Comment 10:

#line 475

Is: “Sample (100 μl; 1 mM [14N]-potassium nitrate) was mixed with 600 μl sulfuric acid of the indicated concentration and 300 μl mesitylene containing 0.5 mM 2,4-dinitrotoluene as internal standard.”

Should be: “Sample (100 μL; 1 mM [14N]-potassium nitrate) was mixed with 600 μL sulfuric acid of the indicated concentration and 300 μL mesitylene containing 0.5 mM 2,4-dinitrotoluene as internal standard.”

Response 10:

The sentence was corrected as suggested.

Comment 11:

#line 485

Is: “Sample (100 μl; 0.5 mM [14N]-potassium nitrate) was mixed with 600 μl sulfuric acid 80% or 85% and 5, 10, 20, 50, 100 or 150 μl mesitylene was added. The mixture was shaken with 1400 rpm at room temperature for 20 min. The reaction was stopped by addition of 500 μl water containing 0.01% indigo carmine. Subsequently, 50 μl 0.5 mM 2,4-dinitrotoluene dissolved in mesitylene was added as internal standard and heptane was added to complete the organic phase to 200 μl.”

Should be: “Sample (100 μL; 0.5 mM [14N]-potassium nitrate) was mixed with 600 μL sulfuric acid 80% or 85% and 5, 10, 20, 50, 100 or 150 μL mesitylene was added. The mixture was shaken with 1400 rpm at room temperature for 20 min. The reaction was stopped by addition of 500 μl water containing 0.01% indigo carmine. Subsequently, 50 μL 0.5 mM 2,4-dinitrotoluene dissolved in mesitylene was added as internal standard and heptane was added to complete the organic phase to 200 μL.”

Response 12:

The paragraph was corrected as suggested.

Comment 13:

#line 493

Is: “Potassium nitrate solutions (100 μl of [14N]-potassium nitrate or 100 μl of [15N]-potassium nitrate) of different concentrations (0, 0.005, 0.01, 0.02, 0.05, 0.1, 0.2, 0.5 and 1 mM) were analyzed as described in 4.4.3.”

Should be: “Potassium nitrate solutions (100 μL of [14N]-potassium nitrate or 100 μL of [15N]-potassium nitrate) of different concentrations (0, 0.005, 0.01, 0.02, 0.05, 0.1, 0.2, 0.5 and 1 mM) were analyzed as described in 4.4.3.”

Response 13:

The sentence was corrected as suggested.

Comment 14:

#line 498:

Is: “A sample was derivatized with 20 μl mesitylene as described in 4.4.3.”

Should be: “A sample was derivatized with 20 μL mesitylene as described in 4.4.3.”

Response 14:

The sentence was corrected as suggested.
